# Prevalence, Related Factors, and Levels of Burnout Syndrome Among Nurses Working in Gynecology and Obstetrics Services: A Systematic Review and Meta-Analysis

**DOI:** 10.3390/ijerph16142585

**Published:** 2019-07-19

**Authors:** Emilia I. De la Fuente-Solana, Nora Suleiman-Martos, Laura Pradas-Hernández, Jose L. Gomez-Urquiza, Guillermo A. Cañadas-De la Fuente, Luis Albendín-García

**Affiliations:** 1Brain, Mind and Behaviour Research Center (CIMCYC), University of Granada, Campus Universitario de Cartuja S.N., 18011 Granada, Spain; 2Faculty of Health Sciences, University of Granada, Calle Cortadura Del Valle S.N., 51001 Ceuta, Spain; 3Andalusian Health Service, Granada. Avenida del Sur N. 11, 18014 Granada, Spain; 4Faculty of Health Sciences, University of Granada, Avenida de la Ilustración N. 60, 18016 Granada, Spain

**Keywords:** burnout, gynecology, meta-analysis, nurses, obstetrics, prevalence

## Abstract

Background: Although burnout levels and the corresponding risk factors have been studied in many nursing services, to date no meta-analytical studies have been undertaken of obstetrics and gynecology units to examine the heterogeneity of burnout in this environment and the variables associated with it. In the present paper, we aim to determine the prevalence, levels, and related factors of burnout syndrome among nurses working in gynecology and obstetrics services. Methods: A systematic review and meta-analysis of the literature were carried out using the following sources: CINAHL (Cumulative Index of Nursing and Allied Health Literature), LILACS (Latin American and Caribbean Health Sciences Literature), Medline, ProQuest (Proquest Health and Medical Complete), SciELO (Scientific Electronic Library Online), and Scopus. Results: Fourteen relevant studies were identified, including, for this meta-analysis, n = 464 nurses. The following prevalence values were obtained: emotional exhaustion 29% (95% CI: 11–52%), depersonalization 19% (95% CI: 6–38%), and low personal accomplishment 44% (95% CI: 18–71%). The burnout variables considered were sociodemographic (age, marital status, number of children, gender), work-related (duration of the workday, nurse-patient ratio, experience or number of miscarriages/abortions), and psychological (anxiety, stress, and verbal violence). Conclusion: Nurses working in obstetrics and gynecology units present high levels of burnout syndrome. In over 33% of the study sample, at least two of the burnout dimensions considered are apparent.

## 1. Introduction

Research has shown that occupational responsibilities may compromise workers’ physical and mental health [1]. Among those affected are healthcare personnel, who are particularly vulnerable to disorders, such as anxiety [2], depression [3], and burnout syndrome [4].

Burnout occurs when the worker is exposed to a series of chronic stressors, which provoke a deterioration in one or more of the dimensions identified [5]. A commonly-used means of determining the magnitude of burnout is the Maslach Burnout Inventory (MBI) questionnaire, which considers three psychological dimensions of the syndrome: emotional exhaustion (EE), caused initially by the sensation of physical over-exertion, which, in turn, generates emotional weariness and loss of interest in the patient; depersonalization (D), characterized by detachment, coldness, cynicism, and indifference; low personal accomplishment (PA), i.e., a negative attitude towards the work, low self-esteem, and lack of job satisfaction, which, in turn, provoke a loss of interest and impaired professional performance [6,7].

The development of burnout syndrome is influenced by many factors, which can be analyzed to identify possible risk profiles. In this respect, relevant sociodemographic factors include the subject’s age, gender, marital status, and the number of children [8,9]. Also, the psychological factors, such as personality type, anxiety, stress, or depression [10], and work-related factors, including a perceived lack of autonomy, the nature of the work environment, salary, and shift work obligation and duration [11,12,13,14], are important. Among the latter aspects, the specific characteristics of the workplace also exert an important influence. Thus, for nurses, the type of patient, family relationships, and the workload at the hospital unit may all contribute to the appearance of burnout [15].

In attempting to overcome the negative consequences of this syndrome, nurses may neglect their personal and occupational obligations, giving rise to negative attitudes towards their work. Given this consideration, systematic reviews and meta-analyses have examined the prevalence of burnout [16] and its associated risk factors [17,18,19] in different hospital units. However, some services, such as gynecology and obstetrics, have received little research attention in this respect.

The obstetrics and gynecology service, caring for women’s sexual and reproductive health, is considered a particularly sensitive area. It requires a strong sense of vocation and considerable emotional control, as the nurses here are responsible for providing optimal comprehensive care throughout the female life cycle. Information and emotional support are essential to meet the needs of mothers, during pregnancy [20], in each stage of childbirth and post-partum [21].

As stated above, the gynecology and obstetrics unit has specific characteristics that distinguish it from all others, especially the close emotional contact between the nurses and their patients and the latter’s extreme vulnerability. For this reason, the main aim of this study is to determine and analyze the prevalence, levels, and related factors of burnout syndrome among nurses working in the area of gynecology and obstetrics.

## 2. Material and Methods

### 2.1. Search Strategy 

A systematic review with meta-analysis was carried out, following the guidelines of the Preferred Reporting Items for Systematic Reviews and Meta-Analyses (PRISMA) statement [22] (Appendix A).

The search was carried out in May 2019 by applying the formula: “burnout AND nurs* AND (obstetrics OR gynaecology OR gynecology)” based on the Medical Subject Headings (MeSH) descriptors. The following sources were consulted: CINAHL (Cumulative Index of Nursing and Allied Health Literature), LILACS (Latin American and Caribbean Health Sciences Literature), Medline, Proquest Platform (Proquest Health and Medical Complete), SciELO (Scientific Electronic Library Online), and Scopus.

### 2.2. Study Selection, Critical Review, and Level of Evidence

In selecting the articles for analysis, the following inclusion criteria were followed: (a) original primary sources; (b) gynecology and obstetrics area; (c) exclusive sample of nurses; (d) published in English, Spanish, Portuguese, or French; (e) no restriction by year of publication; (f) assessment of the level of burnout; (g) outcome measures evaluated by an instrument measuring the level of burnout (MBI, which is based on the triad EE, D, and low PA, and PRoQOL (Professional Quality of Life), which evaluates the quality of working life, including one dimension, for burnout); (h) data on prevalence of burnout. Articles based on mixed samples with other healthcare categories and lacking independent data for gynecology and obstetrics nursing were excluded, as were those with insufficient statistical information.

Two authors selected the articles, according to the abstract and title provided. After removing duplicate articles, the full-text article was consulted. After ensuring compliance with the inclusion criteria, each study was assessed independently, and a consensus was reached between the same two authors regarding the quality of the article. In their evaluation, a checklist was applied to determine the presence/absence of methodological bias. If in any case, the two authors were unable to agree, a third author was consulted.

The methodological quality of each article was assessed using the critical reading checklist proposed by Ciapponi [23]. Specifically, the internal validity of each study was verified by reference to items 1 to 6 and 11 to 18.

The level of evidence was evaluated following the recommendations of the Oxford Centre for Evidence-Based Medicine (OCEBM) Levels of Evidence Working Group [24].

### 2.3. Data Coding

The following study variables were obtained:

Publication variables: (a) authors; (b) year of publication; (c) country; (d) gender distribution (male/female); (e) age; (f) language.

Methodological variables: (g) total sample; (h) type of study; (i) outcome measure (instruments used and measure); (j) original instrument or adapted version; (k) estimated reliability coefficient of the instrument.

Burnout measurement variables: (l) prevalence of high EE, high D, and low PA (the cut-off points for low and high levels of each dimension were applied by the authors of each study depending on the adaptation of the MBI); (m) average or percentage of each dimension; (n) related factors for each dimension.

The data were recorded in a data coding manual by two researchers working independently. Agreement between them was determined by reference to Cohen’s kappa coefficient (mean value: 0.99; minimum: 0.97; maximum: 1) and to the intraclass correlation coefficient (mean value: 0.98; minimum: 0.96; maximum: 1).

### 2.4. Data Analysis 

The data collected in the systematic review were examined by descriptive analysis, in which the information was classified into data tables and categorized accordingly. With the studies that included sufficient statistical information, three random-effects meta-analyses were performed, for the dimensions of high EE, high D, and low PA. The prevalence and the corresponding confidence intervals were analyzed for each dimension.

The publication bias, i.e., the probability of the study being published with statistically significant results, was evaluated by Egger’s linear regression test. Data heterogeneity was assessed using the I^2^ index. All data analysis was performed using the StatsDirect statistical software package (version 3, StatsDirect Ltd., Cambridge, UK).

## 3. Results

In total, fourteen articles were included in this systematic review and meta-analysis. Figure 1 shows the flow diagram of the study selection process. All the studies selected were cross-sectional. Twelve articles (85.71%) measured burnout according to the Maslach Burnout Inventory (MBI) scale, and two were adaptations of this questionnaire. Two articles (14.29%) used the Professional Quality of Life (PRoQOL) questionnaire.

Nine of the studies (64.28%) were conducted in Asia (China, Japan, Korea, Pakistan, Turkey, Saudi Arabia), three (21.43%) in the Americas (Brazil, Mexico, USA), and the rest in Europe and South Africa. Most of the articles (80%) were published between 2012 and 2018. The reliability coefficient of the burnout questionnaire was only estimated in six articles, which reported values ranging from 0.60 to 0.90, considered in every case to be acceptable. Table 1 details these results.

### 3.1. Dimensions of Burnout Syndrome in the Area of Gynecology and Obstetrics

High levels of EE were reported in two studies [32,38], with average scores of 27.59 and 55.8, respectively. Other authors, however, [26,35,37], had observed medium-low levels of EE, with mean scores ranging between 12 and 25.3. In a further three studies, half of the nurses (prevalence: 49.2–52%) presented high levels of EE [27,28,38]. In the final three studies for which these data are available [25,26,34], lower percentages were reported, ranging from 4.9% to 16.3% (Table 2).

For D, high levels were reported, with average scores between 10 and 29.5 [32,35,38]. Other authors had obtained lower mean scores, of five and seven [26,37]. High levels of D were found in 3.8% to 50% of the nurses sampled [25,27,34,38].

Several authors reported low levels of PA, with average scores of 10.9 to 30.06 [32,35,37,38]; only one author [26] reported high levels of PA, with a mean score of 45.1. The prevalence of low PA ranged from 4.8% to 78% of the nurses studied [25,27,38].

About the total score for burnout syndrome, one study [36] reported medium levels, but two others [30,37] recorded low levels among gynecology and obstetrics nurses, with total mean scores of 29.9 and 6.19, respectively. The prevalence of burnout reported in these studies was 0.55% [28], 6.52% [34], 13.1% [33], and 21.4% [25].

### 3.2. Related Factors for Burnout in Gynecology and Obstetrics Services

Among the sociodemographic factors considered, age was found to be a factor related to burnout; thus, lower levels of EE and D were observed in older nurses [25,26,27,35,38]. With respect to gender, women showed higher levels of EE and D, and lower ones of PA than men [27,35,38]. Marital status is another significant factor; single nurses had lower scores for PA than those who are in a stable relationship [25,38]. According to one study, having children is related to higher EE and lower PA [25]. However, two authors found no statistical significance between marital status and number of children, in relation to the risk of burnout [29,34].

Among the work-related factors considered, a work schedule exceeding 48 hours per week was associated with higher levels of EE and D and lower ones of PA [25,26]. The same relationship was observed when the nurses work rotating and/or nocturnal shifts [30,32]. Levels of EE and D were higher with the increase in the patients per nurse ratio [25,33], with the decrease in the number of nurses working in the unit, and among less experienced nurses [25,26,27,29]. However, one study [34] observed no significant relationship between burnout syndrome, experience, and night-time work.

The poor organization was related to higher levels of EE and D [25,28]. Another study [29] concluded that this problem results, fundamentally, in lower PA, while another found that low job satisfaction [36] is a major problem in this respect. Moreover, the gynecology and obstetrics service was reported to be among the units presenting the highest dropout rate in the nursing profession [30]. Finally, low salary levels were also related to lower levels of PA [25].

Other work-related variables that were reported to be significant include working in rural or urban areas [25]; when nurses’ work took place in both areas, the prevalence of burnout was lower. Within the gynecology and obstetrics service, the nurses who provide antenatal care presented lower levels of PA [25].

Some authors [25,32,36] studied the relationship between burnout and the intrinsic characteristics of the work performed in this unit, such as the high degree of involvement by the parents, death, perinatal grief, or participation in a large number of live births or abortions; in the latter respect, the greater the number of first-trimester abortions performed, the greater the risk of burnout, with falling levels of PA, in particular [31].

Finally, in terms of psychological factors, high levels of stress and anxiety were reported, with higher scores for all three dimensions [25,30,36]; on the other hand, the prevalence of depression was low (2.5%) [33]. A negative factor was that of verbal violence, which was associated with increased EE, related to a chronic state of stress [27]. However, 66.4% of the nurses considered their health status to be good [33]. Higher levels of post-traumatic stress were found among older nurses [36], with lower scores for PA.

### 3.3. Levels of Burnout in Comparison with Nurses Working in Other Hospital Services

Regarding the dimension of EE, the hospital areas presenting the highest prevalence and mean scores are those of dialysis, internal and general medicine [29], surgical medicine [33], emergencies and pediatrics [37].

Comparable findings have been reported for D [29]. In this respect, one study reported finding the highest scores for nurses working in pediatrics and obstetrics-gynecology [33], although other authors obtained the highest scores for the emergency, mental health, and surgical medicine areas [37].

Finally, for PA, the results obtained are conflicting. One paper reported that the nurses working in the areas of gynecology and obstetrics were less vulnerable to low PA [29]; others, however, recorded the lowest scores in these same areas, followed by internal medicine and surgery [33,37].

For the three dimensions of burnout as a whole, the intensive care, medical, surgical medicine, and emergency areas have the highest average scores [28], although one study [35] obtained the highest score in this respect for the obstetrics-gynecological area, followed by the surgical medicine, pediatric, and medical areas.

Other studies had observed a greater prevalence of burnout in the areas of pediatrics and surgical medicine [30] or in that of mental health [37].

### 3.4. Results of the Meta-Analysis

The I^2^ index indicated a high level of heterogeneity, with values of 95.7% (95% CI = 93.6–96.9%) for EE, 94.6% (95% CI = 91.5–96.2%) for D, and 97% (95% CI) = 95.8–97.7%) for PA.

The application of Egger’s test of publication bias produced results of 5.52 with p = 0.07 for EE, 4.05 with p = 0.13 for D, and 0.92 with p = 0.92 for PA. We conclude from this that there was no evident publication bias.

In total, 464 nurses working in the gynecology and obstetrics area were included in our meta-analysis. In this sample, the prevalence of high levels of EE was 29% (95% CI = 11–52%) (Figure 2); for D, the value was 19% (95% CI = 6–38%) (Figure 3), and for low PA, the prevalence was 44% (95% CI = 18–71%) (Figure 4).

## 4. Discussion

This study aims to determine the prevalence, level of burnout, and the corresponding related factors in nurses who work in gynecology and obstetrics services, in the view that no previous meta-analyses have been conducted in this respect.

Burnout is commonly experienced by nurses working in these areas. Our analysis shows that the prevalence of high levels of EE is 29%, that of D is 19%, and that of low PA is 44%. These findings are similar to those for nurses working elsewhere, such as the medical area, with high levels of EE among 31% of nurses, of D among 24%, and low levels of PA among 38% [39]; in pediatrics, the corresponding scores are 31%, 21%, and 39% [40]; in primary care, the scores are 28%, 15%, and 31%, respectively [18]; in critical care and emergencies, the scores are 31%, 36%, and 29%, respectively [17].

However, study data reported for other occupational groups working in the same area, such as gynecologists, reflect higher levels of burnout, with 44–56.6% [41,42], among which the prevalence of high EE is 72%, that of high D is 43%, and that of low PA, 74%. Similarly, another study, of medical residents working in the gynecology service, reported high levels of EE and D in 50% of the sample, relating this to a low degree of professional satisfaction and even to regret at having opted for this area of specialization [43].

Regarding the relationship between burnout syndrome and the sociodemographic and work-related variables considered, studies have observed high levels of EE among young people, those who are single, and those with less experience [44] because these groups are less self-confident and are subject to greater tension when decisions must be taken [45]; on the other hand, one study reported that D is positively associated with the number of years spent in the profession [42]. Other researchers have concluded that women are more likely than men to experience burnout [8,46] and that the risk of D is aggravated by women’s greater involvement in care, due to their dual roles, as mother and nurse [47]. Relative youth is also considered a risk factor [8,18] due to these nurses’ greater uncertainty and low expectations of promotion [48]. Besides, it has been reported that PA is negatively correlated with the number of children in the nurse’s family, due to occupational and personal overload [42,49]. On the other hand, more recent studies claim that having children is a protective factor, helping nurses live a fuller life [8].

In terms of organizational characteristics, factors, such as negative sensations regarding the work environment [50], low salaries [51], and lack of organization [52], despite (or in addition to) great responsibility, all contribute to reduced job satisfaction [53] and to nurses’ abandoning the profession [54,55,56]. Also, falling staff numbers and the reorganization of services, due to low birth rates in developed countries [57], increase EE and reduce PA [58]. The lack of resources, concerning the demand for care, tends to make health care highly mechanized and medicalized, a situation in which nurses’ competencies are limited [59], as is their freedom to exercise independent judgment [60]. Furthermore, rotating shifts and the imposition of a 70-hour week increase the risk of burnout, especially in terms of reduced PA [44], due to work overload and heightened levels of stress [61]. Some authors conclude that a lack of commitment, motivation, or time can be considered alarm signals and that the quality of care may be compromised [62,63].

Regarding the psychological variables considered, stress and verbal violence are major risk factors for burnout [64,65] and are related to the high demand for care, the need to attend a large number of patients, and the close link between the patients and the nursing staff [66]. Symptoms of depression are common among these professionals, with a prevalence of 64% [67], which is related to high EE and D and low PA. One of the main characteristics of this hospital area is exposure to traumatic situations. Thus, 25–35% of nurses assisting during births report suffering post-traumatic stress disorders [68,69]. Dealing with loss and alleviating grief is a fundamental aspect of these nurses’ daily work [70], but helping mothers cope with perinatal death (whether natural or resulting from an abortion) increases their vulnerability and the risk of burnout. Possibly, for this reason, levels of PA are lower among nurses working in the antepartum area [71,72].

Finally, regarding the state of perceived health, high levels of EE are related to a high prevalence of physical alterations, such as musculoskeletal injuries [73]. Therefore, many authors show a series of effective interventions that can improve physical and mental health in nurses [74], reducing burnout levels by up to 30% [75]. These include training programs through multidisciplinary workshops, including communication skills, and presenting a positive impact on burnout levels, increasing job satisfaction and improving the level of confidence [76]. In the same way, physical exercise programs, such as yoga, show a reduction in the level of burnout and improve the quality of sleep [77]. Benefits have also been observed in residents of the gynecology and obstetrics area, reducing the blood pressure, as well as the levels of burnout and improvement in their nutritional habits [78]. Other types of interventions are those that increase self-awareness and promote acceptance and motivation towards a change in behavior, such as mindfulness [79], showing a negative correlation between EE, D and positive with PA. Brief 8-week interventions show a reduction in the prevalence of up to 31% in EE [80] and up to 17.60% in D [81], and there is even evidence of an increase in PA levels [82].

### Limitations

Few papers have been published providing sufficient statistical data with which to analyze levels of burnout among gynecology and obstetrics nurses. In consequence, the number of studies included in our meta-analysis is, unfortunately, low. Moreover, the population samples studied are also restricted, and data are not supplied on the number of nurses who have left the profession. All the studies analyzed are cross-sectional; thus, it was not possible to study the long-term impact of burnout on these nurses. In short, only the associations between the variables are analyzed in our study, and the presence or absence of causality is not established. Also, some of the included studies have a small sample, and this must be taken into account before analyzing the related factors. Finally, it must be indicated that a new approach of burnout has established five personal profiles (burnout, engagement, overextended, disengaged, and ineffective) depending on the scores of each burnout dimension [83]. Because the included studies have not used these profiles, we have not been able to establish these profiles.

## 5. Conclusions

Nurses in gynecology and obstetrics services tend to have high levels of EE and D and low levels of PA, but it must be taken into account that due to the influence of occupational variables in each country, these levels may vary.

Some variables that may have a relation with burnout development in gynecological nurses, and that should be analyzed in the future, are being young, relatively inexperienced, single, and/or who have children. Other negative factors in this respect are long working days/weeks and the need to care for large numbers of patients.

Good leadership and appropriate organization of care duties, providing nurses with sufficient autonomy and appropriate staff numbers, are key factors in preventing the development of burnout.

## Figures and Tables

**Figure 1 ijerph-16-02585-f001:**
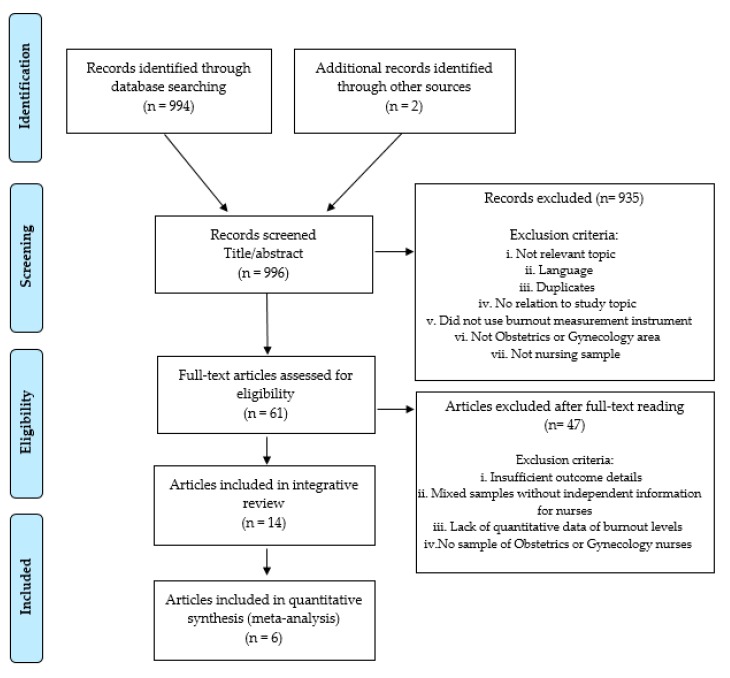
Study selection process to identify eligible articles for inclusion in the review and meta-analysis.

**Figure 2 ijerph-16-02585-f002:**
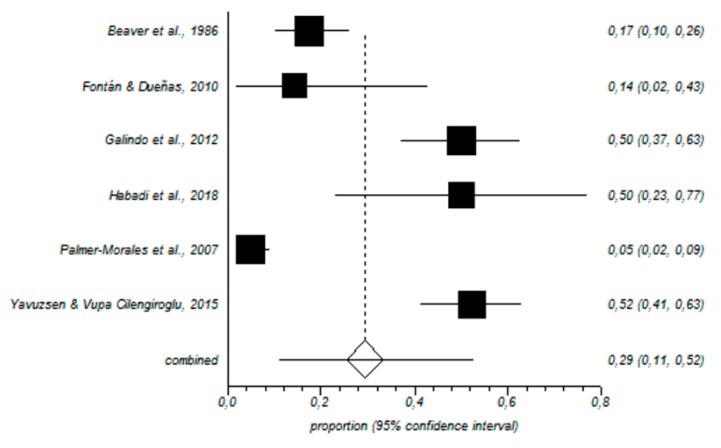
Forest plot for high emotional exhaustion.

**Figure 3 ijerph-16-02585-f003:**
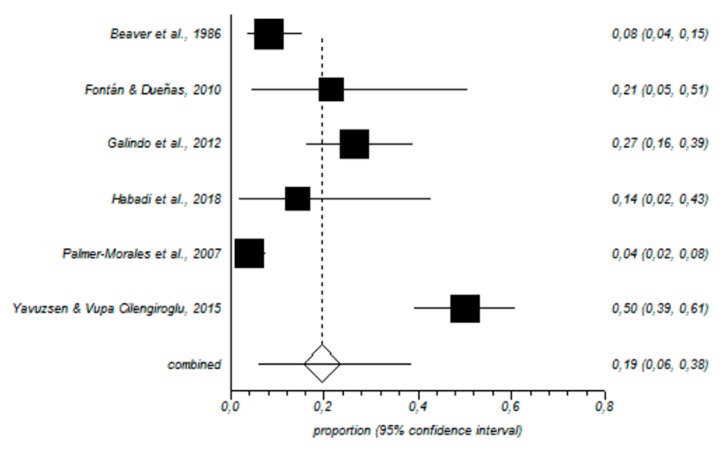
Forest plot for high depersonalization.

**Figure 4 ijerph-16-02585-f004:**
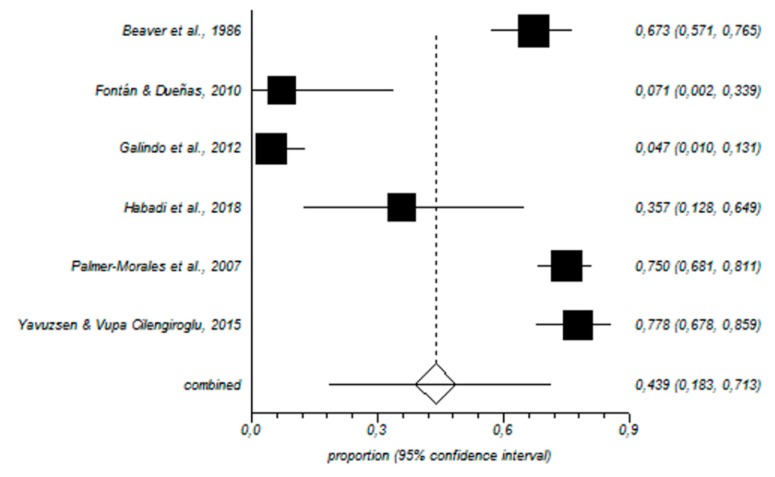
Forest plot for low personal accomplishment.

**Table 1 ijerph-16-02585-t001:** Characteristics of the studies included.

Author, Year, Country	Study Design	Sample (Gender and Mean Age)	Burnout Instrument (Reliability, Cronbach’s α)	M (SD)/Percentage			Main Results	OCEBM GR/LE
				EE	D	PA		
Beaver et al., 1986, USA [25]	Cross-sectional	N = 98 O&G nurses 96.9% Female Age = 30–39: 55.2%	MBI (EE = 0.82, D = 0.60, PA = 0.80)	Low: 58.2% Moderate: 25.5% High: 16.3%	Low: 73.5% Moderate: 18.4% High: 8.2%	Low: 67.3% Moderate: 20.4% High: 12.2%	EE and D are negatively related to age and experience, and positively to the number of births and weekly work hours.	2c/B
Fontán and Dueñas, 2010, Spain [26]	Cross-sectional	N = 14 O&G nurses 78.57% Female Age = 46	MBI	12.2 (10.6) High: 14.2%	5.0 (5.5) High: 21.4%	45.1 (7.1) Low 7.1%	Highest levels are found in professionals who work more than 48 hours per week. Lower level of burnout at older age.	2c/B
Galindo et al., 2012, Brazil [27]	Cross-sectional	N = 64 O&G nurses 92.1% Female Age = 29	MBI (EE = 0.86, D = 0.69, PA = 0.76)	Low: 20.6% Moderate: 30.2% High: 49.2%	Low:14.3% Moderate: 58.7% High: 27%	Low: 4.8% Moderate: 11.1% High: 84.1%	Burnout correlates negatively with salary, experience, and age. A good organization of the service reduces the risk of burnout syndrome.	2c/B
Habadi et al., 2018, Saudi Arabia [28]	Cross-sectional	N = 14 O&G nurses	MBI	High: 50%	High: 14.28%	Low: 28.57%	O&G area is considered one of the lowest prevalences of burnout.	2c/B
Higashiguchi et al., 1999, Japan [29]	Cross-sectional	N = 28 O&G nurses	MBI (Japanese Version)	3.48 (1.29)	1.67 (0.72)	3.64 (1.10)	Low prevalence of burnout with high levels of PA in O&G unit nurses.	2c/B
Liu et al., 2018, China [30]	Cross-sectional	N = 93 O&G nurses	MBI (EE = 0.83, D = 0.83, PA = 0.81)	-	-	-	Low burnout score in O&G nurses (M: 6.19, SD: 2.71). Positive correlation between burnout and rotating shifts (r = 0.444).	2c/B
Mizuno et al., 2013, Japan [31]	Cross-sectional	N = 169 O&G nurses Age = 42.4	ProQOL FEWS	**Subscale**			High emotional burden on nurses in this area. Burnout correlates with the number of abortions, increasing stress, and reducing job satisfaction.	2c/B
				Compassion satisfaction = 33 (6.9) Burnout = 26.75 (5.4) Compassion fatigue = 20.75 (5.65)				
Naz et al., 2016, Pakistan [32]	Cross-sectional	N = 28 O&G nurses	MBI	55.8 (6.7)	29.5 (3.4)	21.8 (4.9)	O&G service nurses have a higher burnout score compared to other services, such as medicine, surgery, neurology, or psychiatry.	2c/B
Nguyen et al., 2018, Korea [33]	Cross-sectional	N = 122 O&G nurses	MBI (Vietnamese version) (EE = 0.89, D = 0.77, PA = 0.80)	2.98 (1.00)	2.72 (0.88)	3.77 (0.77)	Higher EE scores in pediatric and medical area. Higher scores of D and lower in PA in pediatric and O&G area.	2c/B
Palmer-Morales et al., 2007, Mexico [34]	Cross-sectional	N=184 O&G nurses	MBI	Low: 78.8% Moderate: 16.3% High: 4.9%	Low: 91.85% Moderate: 4.35% High: 3.8%	Low: 75% Moderate: 14.7% High: 10.3%	There is no correlation between marital status and number of children and years of work experience with risk of burnout.	2c/B
Sun et al., 1996, China [35]	Cross-sectional	N = 273 O&G nurses	MBI (EE = 0.87, D = 0.81, PA = 0.84)	25.30 (2.99)	12.93 (1.75)	29.90 (2.65)	O&G units present high burnout. The main factor is stress and urgency related to the life of the mother or child.	2c/B
Teffo et al., 2018, South Africa [36]	Cross-sectional	N = 73 O&G nurses	PRoQOL	**Subscale**			An adequate work environment increases motivation and job satisfaction. Burnout is related to years of experience.	2c/B
				Compassion satisfaction = 41 (5.7) Burnout = 33 (4.1) Secondary traumatic stress = 24 (7)				
Yao et al., 2018, China [37]	Cross-sectional	N = 95 O&G nurses	MBI	12.0 (5.9)	7.0 (4.7)	10.9 (8.9)	Emergencies, mental health, and pediatrics are the areas with the highest burnout score. O&G area presents the lowest score in burnout.	2c/B
Yavuzşen and Vupa Çilengiroğlu, 2015, Turkey [38]	Cross-sectional	N = 90 O&G nurses 100% Female Age = 35.49	MBI (EE = 0.90, D = 0.77, PA = 0.74)	27.59 (7.27) High: 52%	10.00 (3.59) High: 50%	30.06 (4.41) Low 78%	D correlates negatively with age. Age, being a woman, and being single are considered related factors.	2c/B

Note: D = Depersonalization; EE = Emotional exhaustion; FEWS = Frankfurt Emotional Work Scale; GR = Grade of recommendation; LE = Level of evidence; MBI = Maslach Burnout Inventory; O&G = Obstetrics and Gynecology; OCEBM = Levels of evidence of the Oxford Centre for Evidence-Based Medicine; PA = Personal accomplishment; PRoQOL = Professional Quality of Life.

**Table 2 ijerph-16-02585-t002:** Prevalence of high EE, high D, and low PA.

Study	Sample Size (n)	High EE (%)	High D (%)	Low PA (%)
Beaver et al., 1986 [25]	98	16.3	8.2	67.3
Fontán & Dueñas, 2010 [26]	14	14.2	21.4	7.1
Galindo et al., 2012 [27]	64	49.2	27	4.8
Habadi et al., 2018 [28]	14	50	14.28	28.57
Palmer-Morales et al., 2007 [34]	184	4.9	3.8	75
Yavuzşen & Vupa Çilengiroğlu, 2015 [38]	90	52	50	78

D = Depersonalization; EE = Emotional exhaustion; PA = Personal accomplishment.

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
