# Peer review of "Prevalence, Related Factors, and Levels of Burnout Syndrome Among Nurses Working in Gynecology and Obstetrics Services: A Systematic Review and Meta-Analysis"

_ijerph, 2019, doi:10.3390/ijerph16142585_

Round 1

Reviewer 1 Report

The paper focuses on burnout and refers in particular to gynaecology and obstetrics services. 

Burnout is one of the most studied variables in healthcare and there is established literature on it. Since the World Health Organization has included burnout in the revision of the ICD-11 as an occupational phenomenon, this research topic is still current and interesting.

Even though all items for systematic reviews and meta-analyses has been observed, please discuss the practical implications of your research more deeply (e.g. consideration on prevention and interventions).

Finally, please make the decimal separator uniform, because you use “,” in the figures and “.” in the text and the tables.

Author Response

Dear Reviewer,

Thank you very much for reviewing the manuscript and your recommendations for improving it. Please find below and highlighted in yellow the response to each recommendation. All the changes in the manuscript have also been highlighted in yellow.

Point 1: The paper focuses on burnout and refers in particular to gynaecology and obstetrics services. Burnout is one of the most studied variables in healthcare and there is established literature on it. Since the World Health Organization has included burnout in the revision of the ICD-11 as an occupational phenomenon, this research topic is still current and interesting.

Even though all items for systematic reviews and meta-analyses has been observed, please discuss the practical implications of your research more deeply (e.g. consideration on prevention and interventions).

Response 1: More information about practical implications have been included in the discussion.

Point 2: Finally, please make the decimal separator uniform, because you use “,” in the figures and “.” in the text and the tables.

Response 2: Dear reviewer, we know that using “.” as decimal separator is the correct form but the software does not allow us to change the decimal separator of the figures.

Reviewer 2 Report

This paper presents a systematic review of burnout and its risk factors in obstetrics and gynaecology. It also presents a meta-analysis on the scale of burnout or burnout subscales in these specialities. The study is presented well and very accessible. The meta-analysis is interesting because it does provide a comparison of sorts however a hierarchy of burnout risk is not usually deemed appropriate, though various authorities take a view that staff in some areas are actually more at risk than others on the basis of clinical intensity. A difficulty is that work stress is highly context-sensitive, not only by clinical speciality but by the work environment at the setting. The analysis of risk seems to me to highlight the likely influence of contextual factors on the variability of the outcomes. Placing obstetrics and gynaecology in clinical context is useful but work factors are less consistent or predictable, and less speciality-specific, than are clinical protocols and outcomes. To summarise my thoughts as to how this manuscript might be improved:

- In the literature search I notice that 'gynaecology' is spelt in a more European way - was 'gynecology' tried? The US publications may be more likely to use this spelling.

- Values for EE, D and PA are presented (p7, lines 140-145) with scores declared 'high', 'low' etc. If cut-off scores were applied then these should be stated. However, the latest user manual for the MBI (Vers 4) does not provide cut-offs on the basis that, according to the designers,  the tool cannot be considered diagnostic in light of new work eg Leiter & Maslach 2016 Burnout Research, 3, 89-100. This should be acknowledged.

- I am not clear as to the selection of the risk factors. A range of factors, both demographic and workplace, are identified but without theoretical underpinning. The MBI is a tool primarily used to evaluate risk of work-related burnout and various models suggest involvement of work environment dimensions that are not included here. This becomes apparent in the Conclusions which cites, amongst other things, autonomy as being preventative yet I couldn't find evidence in the findings to support that.

- I am not clear as to what the message is from this review. The findings highlight variability in most instances between the studies leading to a Discussion reinforced in some instances by findings from just one or two studies. The breadth of potential contributory factors may well explain the variability here, also shown graphically in Figs 2-4. This should be considered in interpreting the data since claims in the Conclusion towards high levels of EE and D are not  especially convincing from the set of data presented. Similarly, the Limitations section (p11) hints at variability in terms of inability to establish 'causality', which may not be feasible anyway in view of the variability of contexts and lack of inclusion of significant work dimensions such as management and peer support. 

Author Response

Response to Reviewer 2 Comments

Dear Reviewer,

Thank you very much for reviewing the manuscript and your recommendations for improving it. Please find below and highlighted in yellow the response to each recommendation. All the changes in the manuscript have also been highlighted in yellow.

This paper presents a systematic review of burnout and its risk factors in obstetrics and gynaecology. It also presents a meta-analysis on the scale of burnout or burnout subscales in these specialities. The study is presented well and very accessible. The meta-analysis is interesting because it does provide a comparison of sorts however a hierarchy of burnout risk is not usually deemed appropriate, though various authorities take a view that staff in some areas are actually more at risk than others on the basis of clinical intensity. A difficulty is that work stress is highly context-sensitive, not only by clinical speciality but by the work environment at the setting. The analysis of risk seems to me to highlight the likely influence of contextual factors on the variability of the outcomes. Placing obstetrics and gynaecology in clinical context is useful but work factors are less consistent or predictable, and less speciality-specific, than are clinical protocols and outcomes. To summarise my thoughts as to how this manuscript might be improved:

Point 1:  In the literature search I notice that 'gynaecology' is spelt in a more European way - was 'gynecology' tried? The US publications may be more likely to use this spelling.

Response 1: The term “gynecology” is also correct. Because “gynecology” is a synonym of “gynaecology” we have tested the search including the term “gynecology” in the search equation "(obstetric OR gynaecology OR gynecology)" and the results of the databases have been the same. The descriptor has been included in the search equation to avoid the same doubt in future readers.

Point 2: Values for EE, D and PA are presented (p7, lines 140-145) with scores declared 'high', 'low' etc. If cut-off scores were applied then these should be stated. However, the latest user manual for the MBI (Vers 4) does not provide cut-offs on the basis that, according to the designers,  the tool cannot be considered diagnostic in light of new work eg Leiter & Maslach 2016 Burnout Research, 3, 89-100. This should be acknowledged

Response 2: We have not established the cut-off points of high, medium or low levels. For the meta-analysis, we have used studies that reported prevalence of high, medium or low levels. The authors of the studies were the ones who applied the cut-off points according to the adaptation of the MBI to their population. We have specified it in the method section and we have included information in the limitations about the 5 profiles established by Leiter & Maslach based on the scores of each dimension.

Point 3:  I am not clear as to the selection of the risk factors. A range of factors, both demographic and workplace, are identified but without theoretical underpinning. The MBI is a tool primarily used to evaluate risk of work-related burnout and various models suggest involvement of work environment dimensions that are not included here. This becomes apparent in the Conclusions which cites, amongst other things, autonomy as being preventative yet I couldn't find evidence in the findings to support that.

Response 3: We have included studies about burnout levels and burnout risk factors in gynecology nurses. The conclusions have been reviewed and modified to avoid establishing statements about the risk factors not supported by the bibliography.

Point 4:  I am not clear as to what the message is from this review. The findings highlight variability in most instances between the studies leading to a Discussion reinforced in some instances by findings from just one or two studies. The breadth of potential contributory factors may well explain the variability here, also shown graphically in Figs 2-4. This should be considered in interpreting the data since claims in the Conclusion towards high levels of EE and D are not  especially convincing from the set of data presented. Similarly, the Limitations section (p11) hints at variability in terms of inability to establish 'causality', which may not be feasible anyway in view of the variability of contexts and lack of inclusion of significant work dimensions such as management and peer support.

Response 4: This review with meta-analysis aims to collect and show the risk factors analyzed in the literature, the existing levels of burnout and to estimate a prevalence of high levels of CE and D and low levels of PA. Following the recommendations of the reviewer because of the variability we have softened and changed the statements related to burnout and their risk factors in the text.

Reviewer 3 Report

The study is relevant and well drafted. However, I would like to reflect on some aspects:

Title and objective: to review "risk factors", since only cross sectional studies were evaluated in which outcome and exposure are measured at the same time, and it is difficult to identify causal relationships between variables. Usually the term risk factors is more appropriate for follow-up studies.

Pubmed and Scielo are not databases, the first is the MEDLINE search engine and scielo is a library.

To clarify the inclusion of studies using different forms of evaluation of burnout, the MBI is based on the triad exhaustion, depersonalization and low professional efficacy, while PROQOL evaluates the quality of working life and one of the dimensions is burnout.

To clarify whether the included studies used the same cutoff point to determine high, medium and low categorization, as well as to identify the prevalence of burnout.

Regarding the factors associated with burnout in this population, some studies indicated association in relatively small samples. In this sense, the authors thought about the possibility of type II error?

Author Response

Response to Reviewer 3 Comments

Dear Reviewer,

Thank you very much for reviewing the manuscript and your recommendations for improving it. Please find below and highlighted in yellow the response to each recommendation. All the changes in the manuscript have also been highlighted in yellow.

Point 1: Title and objective: to review "risk factors", since only cross sectional studies were evaluated in which outcome and exposure are measured at the same time, and it is difficult to identify causal relationships between variables. Usually the term risk factors is more appropriate for follow-up studies.

Response 1: Following your recommendation the term has been changed to “related” factors.

Point 2: Pubmed and Scielo are not databases, the first is the MEDLINE search engine and scielo is a library

Response 2: Thank you for your recommendation. We have modified that information.

Point 3: To clarify the inclusion of studies using different forms of evaluation of burnout, the MBI is based on the triad exhaustion, depersonalization and low professional efficacy, while PROQOL evaluates the quality of working life and one of the dimensions is burnout

Response 3: The information has been included in the method section to clarify it.

Point 4: To clarify whether the included studies used the same cutoff point to determine high, medium and low categorization, as well as to identify the prevalence of burnout

Response 4: The cutoff points were applied by the authors of each study depending on the MBI adaptation for their country. It has been clarified in the methods section.

Point 5: Regarding the factors associated with burnout in this population, some studies indicated association in relatively small samples. In this sense, the authors thought about the possibility of type II error?

Response 5: The quality of the studies was controlled through critical reading. Even so, and due to the low sample of some studies, this information has been included in the limitations of the study.

Round 2

Reviewer 2 Report

I suggest acceptance.

Thank you.